# The Effects of Internal Electron Donors on MgCl_2_-Supported Ziegler–Natta Catalysts for Isotactic PP

**DOI:** 10.3390/polym16192687

**Published:** 2024-09-24

**Authors:** Bin Li, Huashu Li, Hongfan Hu, Yi Zhou, Guoliang Mao, Shixuan Xin

**Affiliations:** 1Provincial Key Laboratory of Polyolefin New Materials, College of Chemistry & Chemical Engineering, The Northeast Petroleum University, Daqing 163000, China; libin14152@163.com (B.L.); maoguoliang@nepu.edu.cn (G.M.); 2PetroChina Petrochemical Research Institute, PetroChina Company Limited, Beijing 102206, China; lihuashu@petrochina.com.cn (H.L.); huhongfan@petrochina.com.cn (H.H.); zhouyi9@petrochina.com.cn (Y.Z.)

**Keywords:** Ziegler–Natta (Z-N) catalyst, internal electron donor (IED), isotactic propylene (iPP), support

## Abstract

The electron donors (ED) in Ziegler–Natta (Z-N) catalysis are classified as internal electron donors (IED) and external electron donors (EED), and both IED and EED are indispensable components for enhancing the catalytic reactivity and regulating the stereoregularity of polyolefinic materials in a typical industrial Z-N catalytic process. With the intensive research on ED, the Z-N catalyst performances have experienced successive progress in the last few decades. Polypropylenes (PP) as a commodity polyolefin material, especially the isotactic PP (iPP), are produced in multi-billion pounds per annum by utilization of the various IED- and EED-assisted Z-N catalysts systems. In the course of developing Z-N catalysts, the ED constitutes a key component of the content and represents a significant area of future research. In this review, we introduced a concise overview of the functions of IEDs in the generations of Z-N catalyst systems and the widely used IED types (A total of 11 different types of IEDs are encompassed within this study) that have been developed so far. In addition, we focused on the coordination modes of different IEDs in the MgCl_2_-supported Z-N catalyst system and analyzed the effects of different types of IEDs on the PP isotacticity, regioselectivity, hydrogen sensitivity, and briefly introduced the application of environmentally friendly rosinate and salicylate IEDs.

## 1. Introduction

Ziegler–Natta (Z-N) catalysts are a group of polyolefin catalysts mainly used for the production of the vast majority of commodity polymers, including but not limited to polyolefins (polyethylene [PEs], polypropylene [PPs], ethylene-propylene-based elastomer [EPM/EPDM] [1,2,3,4], linear low-density polyethylene [LLDPE] [5,6], PP random copolymer [RCP], high-impact PP [HIPP], etc.), and diene polymers (polybutadiene [PB], polyisoprene [PI], etc.) [7,8,9,10,11,12].

Polypropylene (PP) is one of the most widely used thermoplastic resins due to its excellent performance in aspects of stiffness, tensile strength, and transparency. It can also be recycled, which has allowed PP to be used in a wide variety of fields, like packaging materials, automobiles, agriculture, etc. The current industrial manufacture of polyolefins driven by Z-N catalysts has exceeded 100 million tons per annum. This amount accounts for about three-quarters of all the polyolefins production worldwide each year [13]. Undoubtedly, this modified Z-N catalyst, which started with a series of groundbreaking studies by Carl Ziegler and Giulio Natta in the early 1950s, is still the top current in polyolefin catalysts today. Most of the commercial Z-N catalysts used for the industrial manufacture of PP are highly efficient MgCl_2_-supported catalysts, which were developed into the fifth generation. Compared with the original Z-N catalyst system, the most significant change is the addition of MgCl_2_ as support for immobilizing TiCl_4_ and electron donors (Lewis bases, LB) in order to improve the activity and the stereoregularity of PP. The catalyst component of LB, which is directly bonded to MgCl_2_ support, is called internal electron donor (IED) [14,15,16]. IEDs play a crucial role in the MgCl_2_-supported Z-N catalyst and have a great impact on PP products in terms of catalytic activity, molecular weight distribution (MWD), and stereo/region regularity. Novel IEDs such as benzoate, phthalate, ether, and succinate have been introduced to develop a new generation of catalysts. The Z-N catalyst, which has phthalates as IEDs, is one of the most commercially successful and widely used catalyst systems for PP production. However, phthalates have been subjected to a REACH ban due to being potentially harmful to the environment; recently, environmentally benign IEDs, including salicylate IEDs and rosin acid ester IEDs, have been reported. The electron donor added in during cocatalyst addition is called external electron donor (EED) [17]. EEDs not only balance the leaching out of IEDs but also enhance the stereoregularity of PP by activating isospecific sites and poisoning the non-stereospecific sites. 

The MgCl_2_-supported Z-N catalyst is a heterogeneous system with multi-active metal centers. Due to the inherent complexity of the Z-N catalyst system, it is a challenge for researchers to focus on how those factors, such as the difference of IEDs, the complex coordination behaviors of the Ti active components, and the electron donor on different surfaces of MgCl_2_, impact catalyst performance and the characteristics of PP.

This review chronically analyzed the development of Z-N catalysts, focused on the positive effects of each generation of the modern Z-N catalysts, emphasized all of the IEDs that have previously been mentioned in patents and/or the scientific literature, and organized them according to the IEDs’ functional groups. The characteristics of each type of IED were presented in comprehensive detail, with particular emphasis on the manner in which they influence the course of the propylene polymerization reaction. In conclusion, prospective avenues for further research into the development of eco-friendly and sustainable IEDs were proposed.

## 2. The Different Generations of Ziegler–Natta Catalyst

Since the highly efficient TiCl_4_/IED/MgCl_2_-TEA/EED-type Ziegler-Natta catalyst was discovered in the 1980s, many have studied the interaction and mechanism of IEDs and EEDs extensively, mainly aiming to obtain better catalyst systems [18,19,20,21]. In recent years, the stereoselectivity control of the Z-N catalyst system has been improved with the unremitting efforts of research and the use of the continuously updated ED. However, the exact ED functions in highly efficient supported Z-N catalysts are not only related to the solid catalyst itself but also closely related to the type of ED structural feature, ED-addition method, and polymerization conditions. 

The MgCl_2_-supported Z-N catalyst possesses multiple active centers, and one of its common characteristics is slow catalytic activity decay, and the isotacticity of PP polymer is marginally affected by increasing the melting index (reducing molecular weight). Therefore, it is relatively easy to produce polymers with both a high isotactic index (II%) and a high melt index.

The first generation (G-1) catalyst, which involved 3TiCl_3_·AlCl_3_-AlEt_2_Cl, showed a low activity of 0.8–1.2 kg PP/g.cat and a moderate II% of 90–94% [22]. It is also known as the conventional Z-N catalyst [23]. This type of G-1 catalyst system with TiCl_3_ as the main active component and AlEt_2_Cl as the cocatalyst was most commonly used in industry to produce PP before the mid-1970s. However, the catalyst residue (TiCl_3_ and aluminum compounds) needs to be removed after polymerization, and these deashing processes are costly and time- and energy-consuming.

The second generation (G-2) catalyst was obtained by adding LB as an electron donor (ED) and changing the preparation process based on the conventional catalyst 3TiCl_3_·AlCl_3_-AlEt_2_Cl. The β-TiCl_3_ was reduced by AlEt_2_Cl in inert alkane solvents at 0 °C and then treated with ethers at 70 °C to obtain δ-TiCl_3_·R_2_O [24]. The most significant about this generation of catalysts is replacing the β-TiCl_3_ with δ-TiCl_3_. The characteristic of this catalyst is that its particles are small but the specific surface area is large (100–200 m^2^/g.cat). The Solvay catalyst is a classic type of G-2, which was widely used by PP manufacturers in the 1970s and 1980s. These catalysts resulted in significant improvement in the catalytic activity of 10–15 kg PP/g cat, and stereospecificity reached 96–98%. Compared with the G-1 catalyst, the most significant alterations in the G-2 catalyst are the incorporation of LBs and the transformation of TiCl_3_ crystalline form from β-TiCl_3_ to δ-TiCl_3_, which resulted in a noticeable enhancement in catalyst performance (10–15 vs. 1.2 kg PP/g.cat).

The third generation (G-3) catalyst was synthesized by adding a single ester electron donor (ED) [25]. The catalytic activity and stereoselectivity of PP were greatly improved; however, the hidden titanium does not transform into active species during polymerization, and subsequently, only a small fraction of titanium centers initiates the polymer production. These Ti atoms are usually exposed to the surface, edges, and defects of the crystal, and most of the Ti atoms are embedded inside the crystal, thus losing catalytic activity. Therefore, it is still unable to simplify PP production processes by eliminating the deashing operations. In the early 1960s, it was found that loading transition metal compounds on the support with a high specific surface area could disperse the active components on the support surface to give full play to its active role. In the late 1960s, Montedison Company developed a catalyst with activated MgCl_2_ as the support, which showed high activity in propylene polymerization but poor stereoregularity. It was not until the early 1970s that MgCl_2_-supported catalysts with high activity and stereoselectivity were prepared by adding an appropriate electron donor (benzoate). The incorporation of supports (MgCl_2_) and an ED (benzoate) in the G-3 catalyst systems significantly enhanced the activity and stereoregularity of the Z-N catalysts for PP production, and it represents a giant step of advancement in the evolution of Z-N catalysts. The loading of active components and EDs on supports represents a crucial heterogenization process in modern Z-N catalyst preparation. The remarkable morphology of the carrier, as a consequence of the ‘replication effect’, gave rise to the formation of granular PPs with exceptional stereoregularity control.

The fourth generation (G-4) Z-N catalytic system is characterized by replacing a single ester with bifunctional phthalate as IED and adding an organosilicon compound as EED. The performance of the G-4 catalyst was greatly improved; the activity reached 30–60 kg PP/g.cat, and the isotacticity of PP surpassed 98% [26]. Compared with the G-3 catalyst, the increase in polymerization temperature and H_2_ concentration both increased the catalytic activity. The G-4 Z-N catalysts had a certain breadth on molecular weight distribution (MWD) but achieved good performance on temperature profile and also on H_2_ sensitivity. The G-4 catalysts retained the support MgCl_2_ and LB (EDs) that were used in the G-3 catalysts. Additionally, the utilization of bifunctional aromatic diesters and alkoxysilanes as IEDs and EEDs, respectively, led to a profound enhancement in the catalysts’ performance, with an activity of over 15–25 kg PP/g.cat (approximately twice that of the G-3 catalysts) and an improved stereospecificity in the iPP products (II% of 95–99%). Furthermore, the catalysts exhibited the capability of generating regular spherical particles. The principal reason for the extensive utilization of G-4 catalysts is not solely attributable to their superb activity and stereoregularity; the additional advantages of the G-4 catalysts are capable of producing polymers in the form of spherical pellets. When combined with the addition of stabilizers, these pellets can be processed and employed directly, obviating the necessity of the pelletization process, which is obviously unparallel to the first three generations of catalysts [27,28].

The fifth generation (G-5) catalyst was created by adding 1,3-diether, aryl diethers, sebacic ethers as IED [29,30]. The characteristic of this catalyst is that the catalyst activity is very high without the addition of EEDs and that it has high hydrogen sensitivity. Compared with the G-4 catalyst system, the G-5 Z-N catalyst system is about twice more active and has 20 times the H_2_ sensitivity, respectively, [31].

Based on a series of succinate IED-compounded Z-N catalysts, the typical G-5 Z-N catalyst, which contains single succinates, performs well in controlling polymer properties, such as molecular weight. However, they have not been as successful in covering all polymer grades as the G-4 catalyst is. The characteristic of this type of succinate catalyst is a broad MWD. The main characteristics of different generations of Z-N catalysts are summarized in Table 1. The G-5 catalysts exhibit two distinctive characteristics in comparison to the G-4 catalysts. Firstly, the G-5 catalysts with diethers as IEDs demonstrate the ability to produce a PP with a high II% without the need for EEDs. Secondly, with regard to the molecular structure of the compounds, the IEDs in the G-5 catalysts are all phthalate-free structures.

From a chronic account of the advancement of catalyst development, it can be concluded that each generation of Z-N catalysts played a contributory role in the evolution of modern Z-N catalysts. 

The primary contribution of the G-1 catalysts was the proposal that the transformation of crystal shape was beneficial to the enhancement of catalytic activity.

The incorporation of LBs demonstrated to improve the activity of the catalysts, a phenomenon that was reflected in the G-2 catalysts. 

The G-3 catalysts made an invaluable contribution through the introduction of MgCl_2_ support, which proved to be an effective means of enhancing the activity of the catalysts and a simple yet economic way to regulate the polymer morphology.

Furthermore, the G-4 Z-N catalysts identified the use of sets of IEDs and EEDs as a beneficial strategy for optimizing the performance of the catalysts and demonstrated that the combination of IEDs and EEDs in G-4 catalyst systems can markedly enhance the comprehensive catalyst performances in catalytic efficiency, stereospecificity, and polymer morphology. The combination of employing phthalates IEDs and alkoxysilanes EEDs has been shown to result in catalysts with extraordinary activity and outstanding stereoselectivity. The compositional pattern of modern catalysts has, thus, been defined as TiCl4/MgCl2/IED-AlEt3/EED. 

Thus, the development of the G-5 Z-N catalysts has provided ‘non-phthalate esters’ and ‘diethers’ as environmentally friendly IEDs, which are now pacesetters for the formulation of more sustainable IEDs.

## 3. Internal Electron Donors

Common electron donors are Lewis base (LB), which are organic compounds containing oxygen, nitrogen, phosphorus, and silicon. The IEDs are bonded directly to the surface of MgCl_2_ support, which sustains a stronger impact on PP productivity and other properties as well. The main function of the IED is to improve the orthotropic selectivity of the MgCl_2_-supported catalyst. The Mg atoms and chlorine atoms were found in different side sections on the (110) surface and on the (100) surface of activated MgCl_2_ [32]. TiCl_4_ can coordinate with the (110) and (100) faces of MgCl_2_ in the absence of the IED. However, in the presence of the IED, the IED and TiCl_4_ will compete to select coordination on different side surfaces of MgCl_2_ [33]. Due to the weak acidity of Mg atoms on the surface (110), the IED preferentially binds to Mg on the (110) surface so as to avoid the formation of random active centers after the coordination of TiCl_4_ on this surface and to achieve the purpose of controlling the generation of random active centers. The spatial requirements of the donors are shown to be greater on the (110) surface than on the (100) surface, rationalizing the role of LB in the stereo control of polyolefins [34]. In addition to accelerating the grinding activation of MgCl_2_, IED can affect the dispersion of MgCl_2_ · nOH in TiCl_4_ solution and the recrystallization rate of MgCl_2_ and, thus, affect the micro-crystalline structure and morphology of MgCl_2_ in the process of the chemical preparation of highly efficient supported catalyst [35,36]. Therefore, both monoester, diester, and other IEDs on the (110) surface of MgCl_2_ and the coordination of Mg atoms can also prevent the formation of random active centers, which is the main reason why the IED can improve the isotropy of the polymer morphology. Furthermore, the IED could enhance the active surface area and control the amount and distribution of TiCl_4_ on the supported surface [36,37,38]. It is believed that the role of IEDs in the Z-N catalyst system is indispensable in terms of activity, isotacticity, hydrogen response, and MWD manipulation. In this part, we refer to many reports, categorize IEDs according to functional groups, and summarize the effects of IEDs on the catalyst performance and MWD of PP.

### 3.1. Ester Internal Electron Donors

#### 3.1.1. Monoester IEDs

Monoesters of aromatic carboxylic acid like ethyl benzoate (EB) (Figure 1) are commonly used IEDs in this class. Researchers observed that increasing the ester alkyl group length can improve stereospecificity. Soga et al. believed that EB coordination on the MgCl_2_ (100) crystal surface formed higher isotactic active centers, which restricted the conversion of binuclear or multinuclear titanium species into random active centers by preventing the migration of bridging chlorine bonds [39]. The residual alkyl aluminum reacts with EB, which leads to the formation of random active centers. Potapov et al. used diffuse reflectance infrared spectroscopy (DFRIFT) to study the EB catalyst system and found that the internal electron donor EB was mainly bound to the MgCl_2_ (110) crystal surface in a tetrad coordination way, thus preventing the formation of random active centers [40]. Most research results believe that EB only performs coordination with MgCl_2_ and does not coordinate with TiCl_4_. X-ray photoelectron spectroscopy (XPS) analysis of titanium species on the catalyst shows that the catalyst does not contain the TiCl_4_•IED complex, and EB only adsorbs on MgCl_2_, which indirectly affects the adsorption of TiCl_4_ on MgCl_2_, thus affecting the catalytic effect of the active center [41]. Terano et al. believed that compared with the IED of diesters, EB was only coordinated with MgCl_2_, which was less stable than the coordination of diesters with MgCl_2_ and TiCl_4_, which was easy to be extracted by alkyl aluminum with Lewis acid, resulting in low catalyst activity and the isotactic index of PP products [42].

#### 3.1.2. Diester IEDs

##### Phthalate IEDs

In the early 1980s, supported catalysts with better activity and isospecificity performances were developed, which contain a diester as an IED. They were used in combination with alkoxysilane ED. This type of TiCl_4_/IED/MgCl_2_−AlR_3_/EED catalyst (IED = phthalate ester, EED = alkoxysilane) is the most widely used catalyst system in the current PP industry. Among phthalate IEDs, diisobutyl phthalate (DIBP) and dibutyl phthalate (DNBP) are the most representative IEDs (Figure 1). The characteristics of the phthalate IED are that PP has a medium MWD and high activity. Phthalate IEDs are able to control the amount and distribution of TiCl_4_. By comparing the (110) and (104) of δ-MgCl_2_, the TiCl_4_ likely prefers to bind to the (110) surface with the tetra-coordination mode [43]. Introducing phthalate as IED in TiCl_4_/MgCl_2_ catalyst could significantly change the active site distribution on different lateral faces. 

Researchers have carried out theoretical calculations and experimental studies based on the effect of IEDs on the stereospecificity of catalysts, which can be mainly summarized in two aspects [44]. The explanation for the stereospecific effect of IEDs on catalysts is that they convert poorly isospecific sites into high isospecific ones. Another effect is to restrict specific active sites with IED so that the restricted active sites cannot coordinate with TiCl_4_ and lose polymerization activity. According to some studies, the active sites on the MgCl_2_ surface were classified into two types, namely, unstable and poorly isospecific sites (type-1) and stable and highly isospecific sites (type-2) [23,45,46]. There is an equilibrium between the type-1 and type-2 active sites when no EED exists. The introduction of IED can remove some parts of unstable and non-isospecific sites and establish a new balance between type-1 and type-2 active sites on the new defects formed on the surface of MgCl_2_ [23]. By introducing sterically hindered groups on IED moiety, the poorly isospecific sites can be transformed into high isospecific sites. The steric hindrance of the IED, which coordinates in the nearby active Ti center, is the reason for the formation of isotactic MgCl_2_ sites. According to a study by Bukatov, the propagation rate constants (k_g_) and number of active centers (C_g_) showed that the donors decreased sharply the value of k_g_ for non-stereospecific centers, had no effect on the value of k_g_ for stereospecific centers and increased the fraction of stereospecific centers [47]. It is worth noting that this process can be inhibited by adding a cocatalyst. 

Different from the early preliminary understanding that diesters can only chelate stably on the (110) surface, a periodic DFT study pointed out that diesters adsorb equally on the (100) and (110) MgCl_2_ surfaces and form stable chelating coordination on the (110) surface, so the content of the electron donors is more easily affected by the amount of cocatalyst [48]. The cocatalyst can remove some IED from the surface of the catalyst. Busico et al. concluded that the two surfaces (100) and (110) have different acidity, among which surface (110) is more acidic by experimental and theoretical calculations [49]. 

Stukalov pointed out that the IED can competitively adsorb with TiCl_4_ on MgCl_2_ and the IED is more firmly adsorbed on the MgCl_2_ (110) surface [50]. On the surface of the catalyst containing the IED, TiCl_4_ only adsorbs on the surface of the carrier which is inaccessible to the IED. 

Singh speculated that the main reason for the higher activity of the catalyst containing DIBP was that the DIBP molecules could form a bridging structure between adjacent MgCl_2_ wafers, which greatly increased the stability of the catalyst [51]. The spacing between the adjacent magnesium atoms of the MgCl_2_ (110) crystal surface is about 0.27–0.28 nm, while the spacing between the two oxygen atoms in *ortho*-phthalate is about 0.27 nm, and the spacing between the two oxygen atoms of the *meta-* and *para*-diester exceeds 0.5 nm [21,52]. 

The experimental results evidently prove that only the *ortho*-phthalate as the IED can be an effective catalyst. The spacing of the two oxygen atoms in the 1,3-diethers with the same high polymerization activity is within 0.25–0.33 nm, so it is easy to form complexes with Mg atoms. Similarly, Sobota et al. also found that the distance between oxygen atoms of the two functional groups on the IED is the key to whether it can coordinate with TiCl_4_ or MgCl_2_ [52,53,54]. The esters are coordinated with Ti via carbonyl oxygen, and the reaction between *ortho*-phthalate and TiCl_4_ can form seven-membered rings (Figure 2), but *para*-phthalate can only form linear complexes [55].

In exploring the effect of IED molecular structure on propylene polymerization activity and catalyst stereoselectivity, Kakkonen et al. [56] pointed out that the distance between the two carbonyl oxygen atoms in the diesters is particularly important, which is in agreement with the conclusions presented in the previous paper by Sobota et al. [55]. They concluded that diesters capable of forming seven-membered ring chelates with TiCl_4_ and containing the *cis*-OCC=CCO in their structure of molecules as IED catalysts have high activity in propylene polymerization with a high II% of PP. In addition, increasing the alkyl chain length of the phthalate diesters is associated with an increase in catalyst activity; this might be due to the low dielectric constant of longer alkyl chain phthalates and their inability to react with bulk MgCl_2_.

The benefits of diester IEDs are high catalytic activity and good stereospecificity control, but they must be used in conjunction with EEDs during polymerization. In addition, the use of certain plastic materials containing phthalates in household and childcare articles made of or containing parts made of such materials is prohibited in the European Union because the presence of phthalates poses or may pose risks to human health and the environment in the long term. Thus, many researchers have focused on the discovery of more potent electron donors for environmentally friendly catalysts in recent years.

##### Malonate IEDs

Malonates are also applied as non-phthalate IED for a supported Z-N catalyst system. Among the malonates, α-substituted malonates are the intensively studied types [57,58,59]. A few typical types of α-substituted malonates (Figure 3) are 2,2-diethyl diethylmalonate (DEDEM), 2,2-dibutyl diethylmalonate (DBDEM), 2,2-diisobutyl diethylmalonate (DIBDEM), ethylphenyl diethylmalonate (EPDEM), and benzyl diethylmalonate (BDEM). When malonates containing large α-substituents are used, the activity was found to be higher than those with smaller α-substituents in malonates groups. EPDEM showed the highest activity of 49.3 kg PP/g.cat with the highest II%: 97.8%.

Guo et al. synthesized some non-phthalate diesters using diethyl malonate (DEM) as a template and modified the alpha-position (Figure 4) [60]. When these DEM derivatives were used as IEDs, the steric hindrance of α-substituents, the distance of the carbonyl oxygen atoms, and the coordination ability of metals with these IEDs, all contributed to the catalytic activity. The Ti loading is lower when DEDEM with smaller steric hindrance and shorter carbonyl oxygen atomic distance compared with 1,1-cyclopentanedicarboxylic acid diethyl ester (CPCADEE), which may be attributed to the weaker coordination ability of the former than that of the latter. The catalyst system with these IEDs was investigated for propylene polymerization. It was shown that the activity of CPCADEE IED catalysts and polymer properties were improved by the addition of EED (Activity: 10.2 vs. 3.2 kg PP/g.cat; II%: 86.1% vs. 62.8%).

The Z-N catalyst with malonates exhibits good stereospecificity and activity. Substituents on the malonates played a crucial role in improving the stereoselectivity and catalyst activity. Substituent with larger steric hindrance can increase the activity of the catalyst. The steric hindrance of the substituent and the spacing of the oxygen atoms can affect the titanium content in the catalyst. The shorter oxygen spacing and higher electronegativity make Ti coordination stronger, consistent with higher Ti loading. 

##### Succinate IEDs

The succinate-type IEDs have emerged as unique characteristics as these donors have low hydrogen response in propylene polymerization, so they are particularly suitable for PP with a broader MWD [37]. The high activity of these catalysts is due to the ability of succinate ligands to form bridging bonds between adjacent MgCl_2_ crystal faces. The grade of PP made from this catalyst has good processability with applications like injection molding and pipelines [61,62,63]. Many specific substituted succinate compounds like dibutyl succinate (DBS), diethyl-2,3-diisopropyl succinate (DEDPS), and diisobutyl-2,3-diisopropyl succinate (DBDPS) as IEDs (Figure 5) were reported by BASELL [64]. 

The effect of the electron donor structure of succinates IEDs on catalyst performance is similar to that of diether IEDs (vide infra). Steric hindrance can increase both catalytic activity and isotacticity. Wen et al. [37] synthesized catalysts with four different substitutions on 9,10-dihydroanthracene-9,10-α,β-alkyl succinates as IED (Figure 6). It was found that the structure of the IEDs had a significant effect on the performance of the catalysts. In the absence of an external electron donor, the activity of Cat-2, Cat-3, and Cat-4 is close to each other, about four times that of Cat-1. When diphenyl dimethoxysilane (DPDMS) was used as the EED, the activities of Cat-2 and Cat-4 were the highest, and the sensitivity to hydrogen regulation was excellent. The molecular weight of PP prepared by Cat-4 catalyst was very high, four times that of PP prepared by Cat-2. 

Zhang et al. [65] scrutinized the influence of IEDs composed of *rac/meso*-diethyl 2,3-diisopropylsuccinate (DISE) in different proportions on catalyst performance. The results showed that *rac*-2,3-DISE (58.1kg PP/g.cat.h) was more active than *meso-*2,3-DISE, and the resulting PP had a broader MWD (11.8 vs. 7). The distance between the oxygen atoms of the two carbonyl groups on *rac-*2,3-DISE is larger than that on *meso*-2,3-DISE, and it is easier to bridge the coordination on the (110) and (100) face of MgCl_2_. 

Yang et al. developed spiral-substituted succinate compounds (Figure 7) to investigate the effect of different electron donors on propylene polymerization [66]. The use of an unsaturated heterocyclic succinate compound with unique electronic effects as the IED results in high catalytic activity (51 kg PP/g.cat.h), good sensitivity to H_2_ concentration, and high isotactivity of PP (II% = 99%) and wide MWD (7.6–9.2). Without the addition of the EED, the activity of the catalyst containing the IED was still very high (62 kg PP/g.cat.h, and II% ≥ 95%). However, no further study has been conducted on the effect of geometric isomers of electron donors in succinate with helicoidal substitution on catalyst properties. Spiral-substituted succinate compounds are a type of non-phthalate compound that is environmentally friendly. They may be an alternative to phthalate in the future due to the relatively high activity of succinate catalysts and the wide MWD of the polymer.

The Z-N catalyst with succinate IEDs demonstrates a relatively low hydrogen response. This type of IED is mainly used for PP with a broad MWD. EEDs are frequently incorporated into this catalyst to maintain the stereospecificity of the polymer formed and appropriately selected EEDs were necessary for the tuning the isotacticity of PP. The presence of substituent with larger steric hindrance in a succinate structure can enhance both the activity and stereoselectivity of the catalyst. 

##### 1,3-Diol Ester IEDs

1,3-diol esters are a new type of IEDs that have been used in the synthesis of PP catalysts in the early 21st century. The Z-N catalyst containing 1, 3-diol ester IED has high catalytic activity, and its stereoselectivity can be easily adjusted. It can produce high isotactic PP even without EED. Hydrogen sensitivity is strongly influenced by different substituents. Liu et al. reported IED compounds of diol esters with different structures, as shown in Figure 8 [67]. 

PP produced with catalysts with these 1, 3-diol ester types of IEDs have a high isotactic index and a low melt flow rate as well as a wide MWD. By using DNBP and/or 2,4-pentarediol dibenzoate (PDB) (Figure 8), the polymerization experiments showed that the MFR of PP produced by PDB catalyst was very low, which means that PP has higher molecular weight and melting strength. This kind of PP, with high melting strength and wide molecular weight distribution, is widely used in hot molding, injection molding, coating, and other processes and is especially suitable for the production of foamed objects. Some propylene polymerization data are summarized in Table 2.

Zhao et al. used diol ester and diether IEDs as catalysts components for propylene polymerization [68]. The catalyst maintained a very high catalytic activity (150 kg PP/g.cat) and had excellent stereoselectivity. PP with high isotactic index (II% = 98%) and low ash content (30 ppm) can be obtained in the presence or even absence of EEDs.

Gao et al. synthesized Z-N catalysts with diol dibenzoate (Figure 9) as IEDs and other Z-N catalysts with IEDs, such as benzoate, phthalate, and diether (2-isoamyl-2-isopropy-1,3-dimethoxypropane, IAIPDMP) [69]. Compared with the aspects of polymerization dynamic behavior, activity, hydrogen response, and stereospecificity, the results indicate that the catalyst with 2-isoamyl-2-isopropy-1,3-propandiol dibenzoate (IAIPPDB) as IED has high activity (over 54.3 vs. DNBP 40.3 kg PP/g.cat) and stereospecificity (98.8%) but has poor hydrogen response. The obtained polymer has a relatively wider MWD (7.4). Simultaneously, GPC and DSC were used to analyze the polymers, and it was found that the melting temperature of the whole polymer remained constant when the fraction of the higher isotacticity polymer increased. However, when the fraction of polymer with lower isotacticity decreased, the melting temperature of the polymer was negatively correlated with the polymerization temperature. The possible reason is that there are two types of high isospecific active sites in the Z-N catalyst system with dibenzoate as the IED. The isotacticity of the lower one is greatly affected by polymerization temperature, while that of the other active site is inert to temperature variation.

The Z-N catalyst with 1,3-diol ester IEDs exhibits high catalytic performance without the use of EEDs. Moreover, the stereospecificity and the hydrogen sensitivity of the catalysts can be significantly altered by modifying the molecular structure of the 1,3-diol esters. The position of the substituent in the structure of compounds also exerts an influence on the performance of the catalysts. When the substituent group is positioned in the middle of the structure of the compounds, the hydrogen tuning sensitivity of the catalysts can be enhanced. Conversely, when the substituent group is located on both sides, the activity and stereospecificity of the catalyst are improved [70].

#### 3.1.3. Other Ester IEDs

##### Rosinate IEDs

Rosin acid is a natural renewable bio-resource, which is harmless to human and can meet the eco-friendly requirements of IED. Zhang et al. [71,72] reported several novel bio-derived esters based on rosin acid (Figure 10) as eco-friendly IEDs to prepare MgCl_2_-support Z-N catalysts.

This kind of IED can not only improve the isotacticity of the PP but also obtain high transparency PP without adding nucleating agents (light transmittance 93.8%, haze 4.4%). Cat-5, substituted by triple n-butyl groups, has excellent sensitivity to hydrogen. When the hydrogen pressure increases from 0.03 MPa to 0.15 MPa, the MFR of Cat-1 also increases from 11.34 g/10 min to 189.09 g/10 min.

##### Salicylate IEDs

When using traditional phthalate IEDs, the *O*-phthalate diester residuals in plastics have potential harm to human health. Researchers tried all possible ways to find environmentally benign IEDs that are generally derived from sustainable resources. Salicylic acid is an important component of aspirin and is known for its analgesic properties. Salicylic acid derivatives are also widely used in pesticides and cosmetics. Zhou et al. synthesized a variety of hydrocarbon-substituted salicylic acid esters (Figure 11) as an eco-friendly IED (SIDs) in Z-N catalysts [73]. The results show that the relative content of the high molecular weight active center and the stereoselectivity of the catalyst can be improved by appropriately increasing the volume of SIDs in the catalyst. GPC results showed that two vital features of PP produced with SID and DIBP catalysts: (1) MW produced by SID catalysts (SID 5–9) is lower than that of DIBP catalysts; (2) the MW of PP prepared by each active center in different SID was similar, while the MW of PP prepared by each DIBP center was significantly different, indicating that the active center in the SID catalyst was obviously different from the active center in the DIBP catalyst, and the PP chain produced by SID was easier to terminate than that prepared by DIBP IED.

The effect of a series of sulfonyl-substituted salicylate (Figure 11) as IED (SSID) on the polymerization of propylene is reported in the patents. In the case of cyclohexyldimethoxymethylsilane (CHMMS) as EED, the isotacticity of PP is over 95%. The highest selectivity of the catalyst prepared by SSID-3 was II% = 98.2%. The titanium-loading capacity and catalytic activity of SSID-5 catalysts were the highest, at 4.62% and 38.2 kg PP/g.cat, respectively. However, the effect of steric hindrance of SIDs on the stereoselectivity of polymer and the reactivity of catalyst has not been systematically studied.

According to the findings of Ratanasak [74], the preferred adsorption mode for SID donors is the chelate mode. Due to the steric repulsion, the two carbonyl oxygen atoms allow the ED to form a four-coordination complex with a MgCl_2_ (110) surface that enhances the stability of the chelate mode. The researcher explained the steric hindrance effect of the IED by using the relationship between the adsorption energy of the IED and the activity of the catalyst. Comparing five salicylate esters, it was found that there was a good linear relationship between the adsorption energy of the donor and the activity of the catalyst. When the donor had a large group at R_1_ and R_2_ and a phenyl or tert-butyl group at R_3_, it could provide strong adsorption energy, implying that the donor with the strongest adsorption energy provides the Z-N catalyst with the highest activity.

##### Imide Esters IEDs

Zhuang et al. synthesized a series of imide ester IEDs with different substituents (Figure 12) and prepared the corresponding catalysts [75]. The characteristic of these catalysts is that they produce PP with a broad MWD. The imide ester IEDs contain both C=N bonds and ester bonds in the molecular structure; multiple active centers can be formed in the catalyst. This enables PP to maintain high modulus and high impact strength while having good melt fluidity. It is found that the polymerization activity of the catalyst can be improved when the substituents are electron-donating groups, and the stereoregularity of catalysts can be improved with bulkier substituents.

### 3.2. Diether IEDs

With the discovery of 2,2-disubstituents-1,3-dimethoxypropane (1,3-diether), a novel generation of Z-N catalyst was developed. The unique characteristic of diether catalysts is that they have a very high sensitivity to hydrogen, so it is easy to obtain a high melt index polymer and to give PP with narrow MWD and high isotacticity [38,76,77]. Another advantage of a diether electron donor is that it has a strong coordination ability with MgCl_2_ supports and does not react with TiCl_4_ further in the process of catalyst synthesis [78]. It is not easy to replace with AlR_3_ in the process of polymerization and does not react with Ti–C bonds, Ti-H bonds, and Al-C bonds [79]. The Z-N catalysts with 1,3-diethers IEDs are in great demand in the industry due to their improved chemical stability toward TiCl_4_ and cocatalyst (TEA) and are unable to extract these donor molecules from the solid component of the catalyst [80]. PP generated by this catalyst has a good physical, organoleptic appearance and better transparency. It has also been widely used in fiber, healthcare products, and packaging materials. The effects of these donors on PP have been reported in various academic and patent literature [81,82,83].

Among the diether IEDs, substituted 1,3-diether is preferred. The substituent can be alkyl or aryl groups. The 9,9-bis(methoxymethyl)fluorine (BMF, Figure 12) is the most representative IED, and the general structure is as follows (Figure 13):

Many research studies have been conducted to address the relationship between the steric hindrance effect of diethers and polymerization performance in the last two decades [84,85,86]. Thus, when R_1_ and R_4_ are less bulky alkyl substituents in 2-substituted diether donors (Figure 13), it is beneficial for oxygen to interact with the support or active center, and this can improve the isotactic index of PP. Obviously, by comparing the isotacticity of MgCl_2_/TiCl_4_/CBB (CBB=1,1-bis(methoxymethyl)cyclobutene) and MgCl_2_/TiCl_4_/BMF catalysts, it is clear that the stereospecificity of catalyst changes significantly as a function of the BMF/Mg molar ratio. The authors point out that the stereospecificity of the catalyst may be influenced by the structure of the electron donor [79]. BMF has a strong effect on the active site due to the appropriate bulkiness and strong electron-donating groups. Moreover, cyclobutyl in CBB has a weak electron-donating ability and poor steric hindrance, which means that CBB can hardly and effectively exert influence on the stereospecificity of adjacent active sites that coordinate with Mg ions.

Morini et al. prepared three kinds of diethers with different structures and studied their effects on catalyst properties (Figure 14) [87]. The substituents on the 2,2-diisobutyl-1,3-dimethoxypropane (DIBDMP) and 2,2-dicyclopentyl-1,3-dimethoxypropane (DCPDMP) molecules are bulkier groups, while 2-ethyl-2-butyl-1,3-dimethoxypropane (EBDMP) are linear and less bulky. The results showed that the yield and isotacticity of PP from DIBDMP and DCPDMP catalysts were both higher than those of Z-N catalysts with EBDMP. Compared with the MgCl_2_/TiCl_4_/DIBP-TEA/phenyltriethoxysilane (PTES) system, catalysts containing DIBDMP and DCPDMP have high productivity of PP, and the isotacticity of PP is almost the same.

The impact of diethers on the polymerization process at the molecular level was investigated by Morini and colleagues [87]. They employed TREF (Temperature Rising Elution Fractionation) and ^13^C-NMR to analyze the stereo sequence distribution of products generated by the catalysts with DIBDMP, DCPDMP, and EBDMP IEDs. The fractions were categorized into four types (atactic, stereo blocks, mainly isotactic, and highly isotactic) based on their elution temperature ranges. The results demonstrated that the PPs obtained from DIBDMP and DCPDMP IEDs were predominantly the highly isotactic fraction. In comparison with the phthalate/siloxane catalyst system, it was observed that the diether IEDs had the effect of reducing the content of stereo-block fraction in two ways: (a) saturating Mg atoms coordination vacancy adjacent to isospecific dinuclear Ti adduct and (b) saturating the vacancy of inactive Ti of Ti_2_Cl_6_. However, a comparison of EBDMP with the ED-free catalysts revealed that, despite EBDMP showing a higher isotactic index (II% ≈ 92%), approximately 53% of the polymer was fraction 3 (mainly isotactic) and contained minimal fraction 4 (highly isotactic). A comparison of the elution curves and melting points of the polymers revealed that they were similar to those of the catalyst without the ED. The authors depicted two particular reasons for the absence of fraction 4 in EBDMP catalysts: (a) the formation of chelates on Mg atoms is not sufficiently stable, and (b) substituents with less bulky substitution groups are not efficient enough to coordinate on Mg atoms adjacent to active isospecific Ti centers.

The distance between two oxygen atoms in 1,3-diether IEDs has a great influence on the performance of the supported catalyst. Through theoretical calculation and structural analysis [88,89], it can be concluded that (1) the activity and stereotaxis of the catalyst are practically good when the O–O distance is close to 3Å, but the performance of the catalyst is poor when the spacing is wider than 3Å, and (2) steric hindrance around oxygen atoms also has an effect on catalyst performance. When the substituent group on the oxygen atom is replaced by the bulkier groups, the coordination between the diether and the support MgCl_2_ is hindered, resulting in the decrease of catalytic activity and stereoregularity.

Several studies attempted to deduce the correlation between the coordination of diether IEDs on the MgCl_2_ surface and the polymerization performance in the last two decades [72,82,90]. The 1,3-diether IEDs adsorb relatively more easily on the MgCl_2_ (110) surface rather than on the (100) surface, while phthalate IEDs have nearly equal affinity for both (110) and the (100) surfaces [24,78,79]. Kumar Vanka et al. [91] studied the interaction energy between the Mg atom and IED on the MgCl_2_ (110) surface using the DFT calculation, and the results showed that there was chelate and bridge coordination between the diester IEDs and MgCl_2_ crystals. The preferred mode of coordination of 1,3-diether compounds on the MgCl_2_ surface is the chelate mode, which is very effective under the action of AlEt_3_, and these molecules do not leach out during polymerization. Lee et al. [92] used the (MgCl_2_)_n_ clustering model to calculate the adsorption energy barrier between the diether electron donor inserted into the active center of propylene and MgCl_2_ crystal faces. The results show that the adsorption energy of MgCl_2_ (110) and (100) surfaces containing branched structures in the diether IED is greater, and the stereotaxis ability of the catalyst is stronger and more effective. At the same time, it has been found that the effect of diether on MgCl_2_ (110) crystal surface is more reflected in the process of catalyst preparation, reducing the number of random active centers on the surface of MgCl_2_ (110) crystal surface, but does not change the number of random active centers on the crystal surface of the isotactic active centers. Liu’s group concluded that the preferred co-adsorption mode of diether type IEDs is the chelate mode on the 4-coordinated Mg site adjacent to the Ti active site [21]. In the absence of electron donors, the bare active sites showed to be regioselective and non-stereoselective during the polymerization of propylene, which can be attributed to two reasons [78]: (1) 2,1-insertion is less stable than 1,2-insertion due to its higher apparent activation energy on the surface of MgCl_2_; (2) there is steric hindrance between the methyl group of propylene monomer, which leads to the α-CH_3_ of the growing chain preventing the further π-complexation and insertion of propylene.

In summary, the Z-N catalyst with 1,3-diether IEDs exhibits good stereospecificity and a good hydrogen response, as well as an eco-friendly nature, making it of increased industrial importance. PPs produced by this type of IED-containing catalysts have good physical and organoleptic properties and better transparency; the PPs are attractive for applications in fiber, healthcare products, and packaging materials. Substituents on the diether are crucial in improving the stereoselectivity and catalyst activity, as discussed previously.

### 3.3. Other Types of IEDs

#### 3.3.1. IEDs Containing Sulfonyl Groups

The catalyst prepared with disulfonyl IEDs has high activity and stereoselectivity. The disulfonyl IED can be combined with the diester IED to prepare an efficient Z-N catalyst for olefin polymerization and copolymerization, which is characterized by high activity, high stereoselectivity, good hydrogen sensitivity, uniform polymer particles, and low fine content. Li et al. reported several disulfonyl types of IEDs based on the bis(trifluoromethylsulfonyl)phenylamines core structure (Figure 15), which were used for the preparation of Z-N catalysts [93]. Due to its outstanding hydrogen response [94], the melt index of PP changes significantly with hydrogen feed variation. The activity of these catalysts is dramatically influenced by the electronic environment of the phenyl group on the sulfonyl phenylamines. The electron-withdrawing groups on the phenyl ring can enhance both the catalytic activity and the isotacticity in propylene polymerization. A high-performance catalyst with a catalytic activity of 16.9 kg PP/g.cat and an II% higher than 98% can be obtained by combining PTES.

Cui et al. reported in the patent that phosphate-substituted sulfonyl IEDs (Figure 16) in olefin polymerization [95]. Comparing it with the G-4 Z-N catalyst, the catalyst based on these sulfonyl IEDs has more advantages in improving the catalytic activity and isotacticity of the PP. 

#### 3.3.2. IEDs Containing Dibenzoyl Sulfide Groups

Kim et al. synthesized a series of dibenzoyl sulfides IEDs (Figure 17) [96]. Using TiCl_4_/dibenzoyl sulfides IEDs/MgCl_2_-TEA/dicyclopentyl dimethoxysilane (DCPMS) Z-N catalyst for propylene slurry polymerization under atmospheric pressure at a constant temperature of 70 °C, DIBP was used as reference under identical conditions; the results showed that the dibenzoyl sulfide could be an alternative candidate to show the superiority to DIBP in the activity of catalyst (40 vs. 22 kg PP/g.catalyst), the isotacticity of polymer (99.5 vs. 98.0 wt %), and the molecular weight distribution of the PP product (M_w_/M_n_ = 4.8 vs. 4). The authors suggest that Cat-3 not only has sufficiently dispersed active sites evenly on the support surface but also makes the Ti absorption by IEDs easier. Furthermore, in the methyl regions of the PP identified through the ^13^C-NMR spectra, the isotactic sequence lengths (usually presented as ‘mmmm%’ or simply mmmm, meaning that the pentads intensity percentage from the ^13^C NMR spectra of the methyl region signals integral sum) is higher in the PP obtained from Cat-3 than in the PP products from reference DIBP and Cat-2, which can clearly show the IED structural effects on the isotacticity of PP (II%: Cat-3 99.5% > Cat-2: 98.1% > Ref: 98.0%). 

## 4. Concluding Remarks

MgCl_2_-supported Z-N catalysts are still the indispensable catalytic systems for the modern production of commodity polyolefins, especially in isotactic PP and copolymers production. This kind of multi-active center heterogeneous catalyst, which is composed of support, active component, and ED that regulates polymer properties. The Z-N catalysts system is still under intensive investigation and progressive improvement. Each component in the catalyst can be practically adjusted to meet the requirements for the production of the different molecular structures and properties of polymers. 

The two principal factors that influence the activity and stereoselectivity of the catalyst by the IED are as follows: (1) The mode of adsorption of the IED on the MgCl_2_ crystal surface. For instance, this can be observed from the impact of phthalates and diether adsorption modes. The former are subject to the nucleophilic reaction of the A-C bond with the carbonyl group in AlR_3_ during the polymerization process. This results in the removal of these Lewis acids from the MgCl_2_ surface, thereby reducing the active center. In contrast, the latter does not undergo reaction with the Ti-C and Al-C bonds and thus exhibits greater stability upon binding to the MgCl_2_ surface. This results in the catalyst displaying high levels of both activity and stereoselectivity. (2) The molecular structure of the IED. The position of the substituent group in the molecular structure of the IED, the steric hindrance and the electronic effect represent significant influencing factors. It has been observed that the molecular structure of 1,3-diol ester IED, in which the substituent groups are positioned on both sides, has the effect of enhancing both the activity and stereoselectivity of the catalyst. Similarly, the molecular structure of salicylate IED has been observed to enhance catalyst activity when substituent groups with larger steric hindrances are present at R_3_.

At present, phthalate catalysts are still widely used due to their high performance in industrial processes and for economic reasons, but owning to the potential risks to the environment and health, there is also an urgent need to develop non-phthalate Ziegler–Natta catalysts with higher catalytic activity and better stereoselectivity. This context briefly introduces the historical line of the development of Ziegler–Natta catalysts, with emphasis on the chemical structure and application of various types of internal electron donors, such as esters, ether, and other types of IEDs, and their effects on polymer properties. Our intention is to summarize the existing knowledge of IEDs in order to help academic and industrial researchers design new internal electron donor structures that can improve the performance of current Ziegler–Natta catalyst systems.

## Figures and Tables

**Figure 1 polymers-16-02687-f001:**
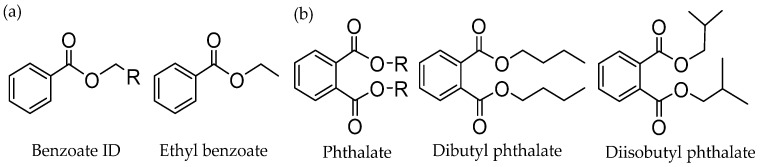
The common types of benzoate (**a**) and phthalate (**b**).

**Figure 2 polymers-16-02687-f002:**
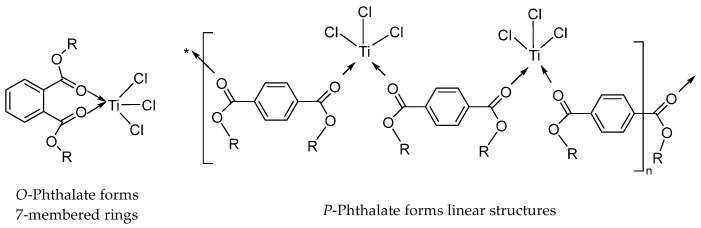
*O*-phthalate forms a seven-membered ring stable structure. *P*-phthalate forms linear structures.

**Figure 3 polymers-16-02687-f003:**
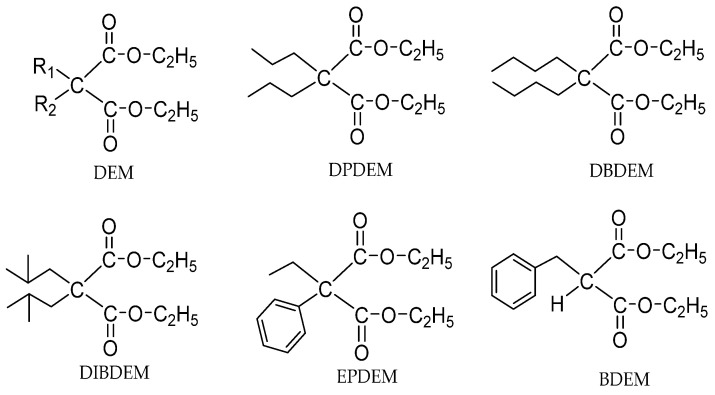
A few typical kinds of α-substituted malonates.

**Figure 4 polymers-16-02687-f004:**
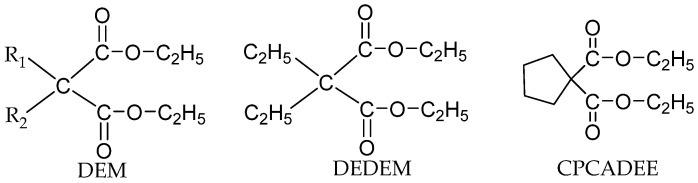
New non-phthalate diesters derive from diethyl malonate.

**Figure 5 polymers-16-02687-f005:**
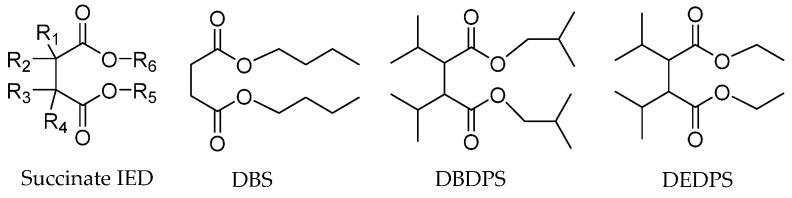
A few examples of BASELL-reported succinate IEDs.

**Figure 6 polymers-16-02687-f006:**
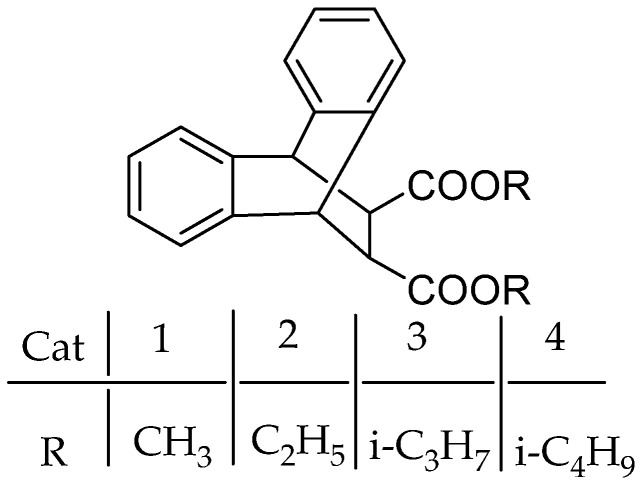
The structures of 9,10-dihydroanthracene-9,10-α, β-alkyl succinates.

**Figure 7 polymers-16-02687-f007:**
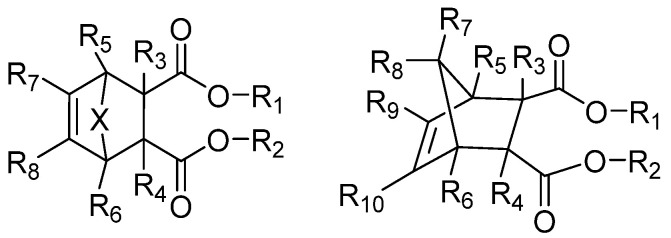
The spiral-substituted succinate compounds.

**Figure 8 polymers-16-02687-f008:**
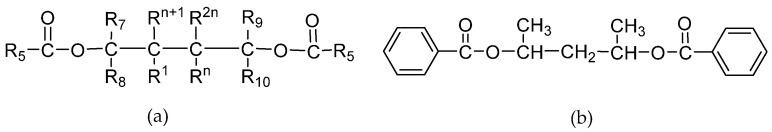
Different structures of diol esters (**a**) and 2,4-pentanediol dibenzoate (**b**).

**Figure 9 polymers-16-02687-f009:**
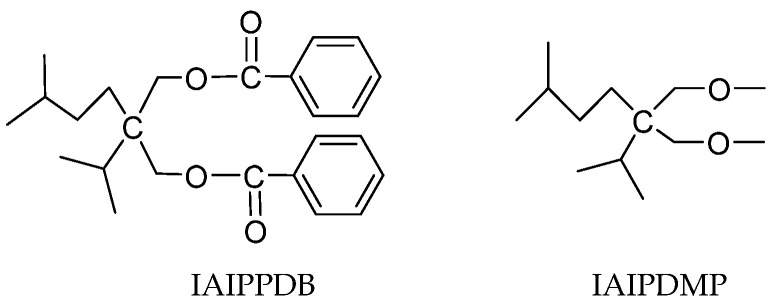
The structure of IAIPPDB and IAIPDMP.

**Figure 10 polymers-16-02687-f010:**
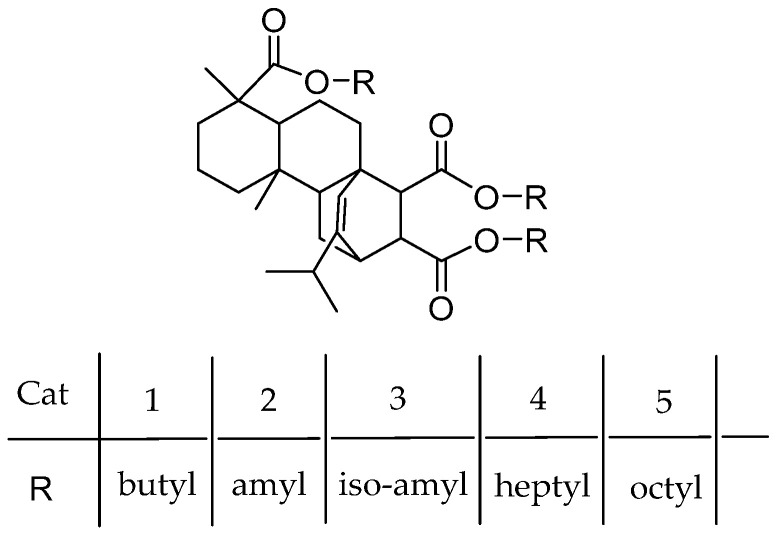
Different substituents of maleic rosinate.

**Figure 11 polymers-16-02687-f011:**
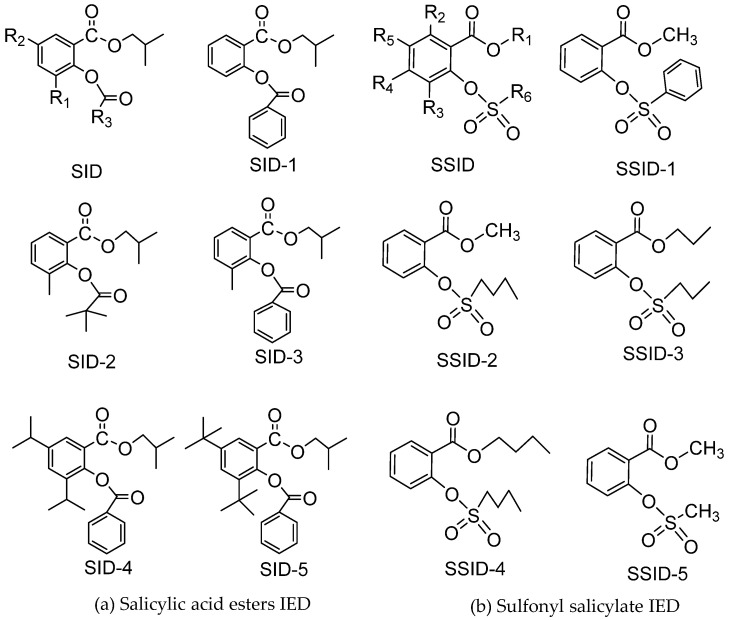
The salicylic acid ester IED (**a**) and sulfonyl salicylate IED (**b**) with different substituents.

**Figure 12 polymers-16-02687-f012:**
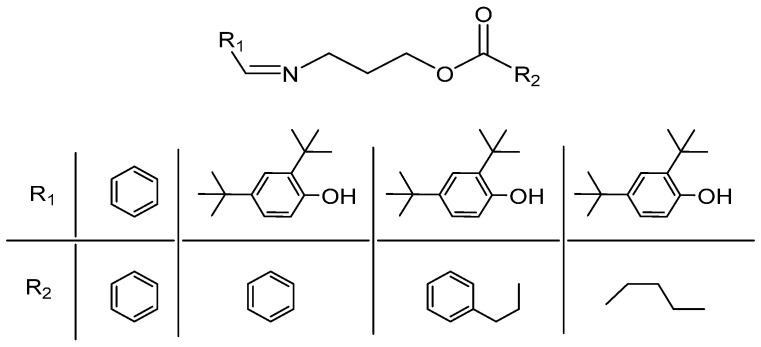
Different structures of imide ester compounds.

**Figure 13 polymers-16-02687-f013:**
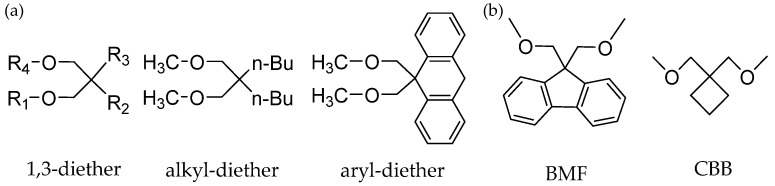
General structure of diether (**a**) and the structure of BMF and CBB (**b**).

**Figure 14 polymers-16-02687-f014:**
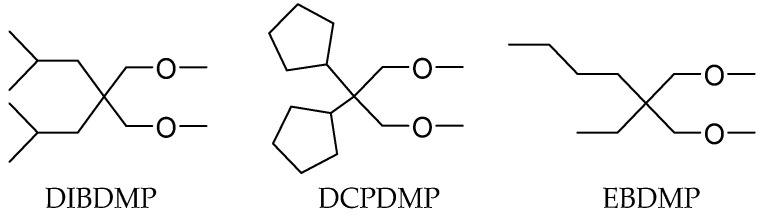
Three kinds of diethers with different backbone bulkiness structures.

**Figure 15 polymers-16-02687-f015:**
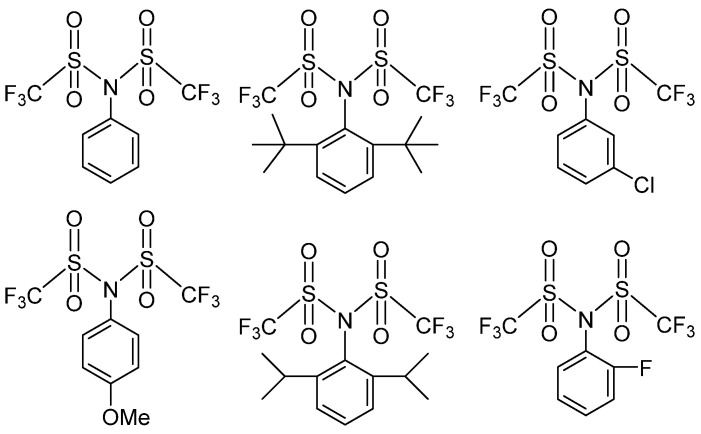
Different structure of bis(trifluoromethylsulfonyl)phenylamines.

**Figure 16 polymers-16-02687-f016:**
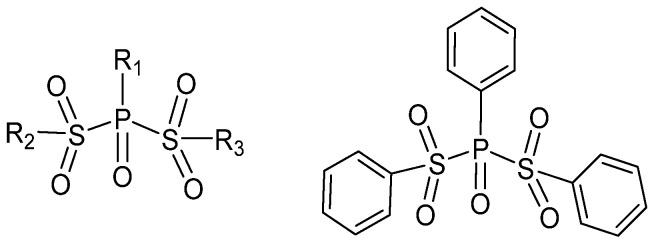
The structure of phosphate-substituted sulfonyl compounds.

**Figure 17 polymers-16-02687-f017:**
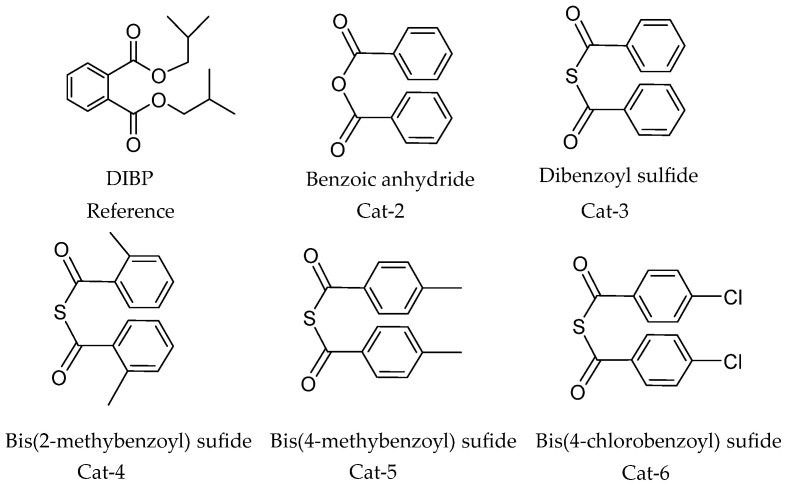
A series of compounds derived from dibenzoyl sulfides.

**Table 1 polymers-16-02687-t001:** Different generations of Z-N catalysts.

Gen	Catalyst Composition	Activity ^a^	II ^b^	Morphological Characteristics	Remakes
1st	3TiCl_3_·AlCl_3_-AlEt_2_Cl	0.8–1.2	90–94	Irregular powder	Need reprocess
2nd	TiCl_3_-AlEt_2_Cl	3–5	96–98	Particle	Deashing catalyst powder
3rd	TiCl_4_/MgCl_2_/Benzoate-AlEt_3_	5–10	95–97	Regular particle	Deashing ataic polymer
4th	TiCl_4_/MgCl_2_/Phthalate-AlEt_3_/Silane	15–25	95–99	Similar Sphere	Nothing operations
5th	TiCl_4_/MgCl_2_/Diether-AlEt_3/_Silane	25–35	95–99	Sphere	Nothing operations
5th	TiCl_4_/MgCl_2_/Succinate-AlEt_3_/Silane	15–20	95–99	Sphere	Nothing operations

Polymerization condition: propylene slurry polymerization, 70 °C, 1 h. ^a^ kg of polymer/g of catalyst. ^b^ Isotactic index (weight percent of heptane insoluble fraction).

**Table 2 polymers-16-02687-t002:** Influence of different IEDs on propylene polymerization.

Catalyst	Internal Donor	Activity ^a^	Ti wt%	II ^b^	MWD	MFR ^c^
Ref	DNBP	42.5	2.3	98.0	5.5	4.5
Cat-1	PDB	56.7	2.4	96.0	7.8	0.8
Cat-2	PDB+DNBP	57.9	2.5	98.3	6.5	1.7

Polymerization condition: C_3_H_6_ = 2.3 L, TEA = 0.05 μmol, CHMDMS = 25 μmol, 70 °C, 1 h. ^a^ kg of PP/g of catalyst. ^b^ Isotactic index (weight percent of heptane insoluble fraction). ^c^ Melt flow rate (g/10 min).

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
