# Peer review of "The Effects of Internal Electron Donors on MgCl2-Supported Ziegler–Natta Catalysts for Isotactic PP"

_polymers, 2024, doi:10.3390/polym16192687_

Round 1
Reviewer 1 Report
Comments and Suggestions for Authors
This review, titled "The Effects of Internal Electron Donors on MgCl2-Supported Ziegler-Natta Catalysts for Isotactic Polypropylene," explores the influence of internal electron donors on MgCl2-supported Ziegler-Natta catalysts in the production of isotactic polypropylene (iPP). It provides a comprehensive examination of the development of Ziegler-Natta catalysts across different generations, emphasizing the changes in catalyst composition and their impact on performance over time. While the review is well-researched and offers a detailed overview of the subject, further clarification and additional insights would enhance its accessibility and impact for a broader audience within the scientific community.
§ The abstract is concise but could benefit from including more specific data points or findings to provide readers with a clearer and deeper overview of the paper’s contributions.
§ The paper begins with an introduction to Ziegler-Natta catalysts, highlighting their significance in the production of polyolefins such as polypropylene. However, it lacks sufficient references, as several broad and general statements are made without proper citation.
§ Although the paper offers a comprehensive overview of various internal electron donors (IEDs), certain sections would benefit from a more in-depth analysis of how these IEDs impact the polymerization process at the molecular level. Incorporating additional data from recent studies or experiments could strengthen the discussion.
§ Could the authors expand the discussion on how each generation of catalysts has contributed to the advancement of modern catalysts?
§ The conclusion could be more robust by providing a more detailed summary of the key findings and discussing potential future research directions in this field.
Comments on the Quality of English LanguageThe English language in certain sections needed more editing and refinement
Reviewer 2 Report
Comments and Suggestions for Authors
This review summarizes the different generations of Z-N catalyst and internal electron donors, the effects of ester-based IED, diester-based IED and other types of IED on PP isotacticity and polymer performance. However, the authors listed references in detail but lacked some systematic organization, analysis and summary. Therefore, I can hardly recommend it for adoption in its current version, the following points make me uneasy.
1.The introduction should indicate what the main points of your review are compared to other similar literature reviews?
2.For Chapter 2, if possible, a summary should be made of what generation of Z-N catalysts are currently used in practice and why?
3.Almost every chapter provides only examples without a summary. It is recommended to summarize the strengths and weaknesses at the end of each section.
4.The [mmmm]..... what is this abbreviation (p.14)
5.Authors should provide some relevant references to draw these conclusions:
-Researchers have carried out theoretical calculations and experimental studies based on the effect of IEDs on the stereospecificity of catalysts, which can be mainly summarized to two aspects (page 5)
-According to some studies, the active sites on MgCl2 surface were classified as two types, namely, unstable and poorly isospecific sites (type-1) and stable and highly isospecific sites (type-2). There is an equilibrium between the type1 and type-2 active sites when there is no EED existed. (page 5)
-It is worth noting that this process can be inhibited by adding a cocatalyst. (page 5)
-It was shown that the activity of CPCADEE catalysts and polymer properties were improved by the addition of EED.(page 7)
-Many research works have been conducted in trying to conclude the relationship between steric-hindrance effect of diethers and polymerization performance in the last two decades. (page 12)
6. In conclusion, please summarize what factors and how can influence the activity of the Z-N catalyst.
7. The authors cite the work of Zhao et al., [ref. 57], but the reference list contains a different name. Also in references [59,60], the authors cite the work of Huang et al., but the reference list contains works by another author. The authors cite the article by Kumar Vanka et al.[80], but the reference list contains works by another author.
Cui et al. [79], but the reference list contains works by another author.
The authors cite the work of Lee et al.[81], but the reference list does not contain this reference.
Reviewer 3 Report
Comments and Suggestions for Authors
The authors conducted a review on the effects of internal electron donors on MgCl2 Supported Ziegler-Natta catalysts for isotactic PP.
1. The review methodology and the trends observed in literature should be included in the abstract.
2. The word "survey" used in the last paragraph of the introduction is not appropriate. In my opinion, the "survey" should be replaced by "review"
3. In the last paragraph of the introduction "...electron donors that have been reported by the researchers over in past decades...""over" should be removed from the sentence.
4. In section 2, "Since the highly efficient TiCl4/IED/MgCl2-TEA/EED type Ziegler-Natta catalyst was discovered in the 1980s, many have studied the interaction and mechanism of IEDs and EEDs extensively, mainly aiming to obtain better catalyst systems." There are no references to back the statement.
5. Also, in section 2, paragraphs 1-4 were not referenced.
6. The authors did not presents a critical review of literature showing how internal electron donors affect MgCl2 Supported Ziegler-Natta catalysts.
7. The review objective is not clear, and it was certain what the authors intend to achieve in the review.
8. Most of the information presented were not properly referenced.
9. There are several language and syntax errors, the authors need to critically revise the manuscript.
Comments on the Quality of English LanguageThe English Language needs to be substantially improved.
Round 2
Reviewer 1 Report
Comments and Suggestions for Authors
The authors addressed fine the comments addressed and improved the quality of the review. I recommend acceptance.
Reviewer 2 Report
Comments and Suggestions for Authors
The authors did not fully respond to the comments of the first review. In the file "authors' responses" I did not find answers to questions 6 and 7
Reviewer 3 Report
Comments and Suggestions for Authors
The manuscript has been properly revised, and the suggestions have been incorporated. I therefore recommend the manuscript for acceptance.
Round 3
Reviewer 2 Report
Comments and Suggestions for Authors
The authors have taken into account all comments and suggestions, the article can be recommended for publication.